

# On the classification of simple and complex biological images using Krawtchouk moments and Generalized pseudo-Zernike moments: a case study with fly wing images and breast cancer mammograms

Jia Yin Goh[1] and Tsung Fei Khang[1,2]

[1] Institute of Mathematical Sciences, Faculty of Science, Universiti Malaya, Kuala Lumpur, Malaysia
[2] Universiti Malaya Centre for Data Analytics, Universiti Malaya, Kuala Lumpur, Malaysia

## ABSTRACT

In image analysis, orthogonal moments are useful mathematical transformations for creating new features from digital images. Moreover, orthogonal moment invariants produce image features that are resistant to translation, rotation, and scaling operations. Here, we show the result of a case study in biological image analysis to help researchers judge the potential efficacy of image features derived from orthogonal moments in a machine learning context. In taxonomic classification of forensically important flies from the Sarcophagidae and the Calliphoridae family ($n = 74$), we found the GUIDE random forests model was able to completely classify samples from 15 different species correctly based on Krawtchouk moment invariant features generated from fly wing images, with zero out-of-bag error probability. For the more challenging problem of classifying breast masses based solely on digital mammograms from the CBIS-DDSM database ($n = 1,151$), we found that image features generated from the Generalized pseudo-Zernike moments and the Krawtchouk moments only enabled the GUIDE kernel model to achieve modest classification performance. However, using the predicted probability of malignancy from GUIDE as a feature together with five expert features resulted in a reasonably good model that has mean sensitivity of 85%, mean specificity of 61%, and mean accuracy of 70%. We conclude that orthogonal moments have high potential as informative image features in taxonomic classification problems where the patterns of biological variations are not overly complex. For more complicated and heterogeneous patterns of biological variations such as those present in medical images, relying on orthogonal moments alone to reach strong classification performance is unrealistic, but integrating prediction result using them with carefully selected expert features may still produce reasonably good prediction models.

Corresponding author
Tsung Fei Khang,
tfkhang@um.edu.my

## INTRODUCTION

Image analysis, the extraction of information from digital pictures by quantitative means, is a powerful way to study biological variation without directly interacting with the physical object that is imaged. After suitable image processing steps such as zooming or reduction, denoising, and segmentation, the pattern of shape variation in an image, which is represented by a matrix of pixel values, can be extracted using suitable feature extraction methods (*Gonzalez & Woods, 2002*). To be practically useful, such methods need to be invariant to translation, rotation, and scaling.

Moment invariants, which are abstract representations of shape that satisfy the three properties of translation, rotation and scale invariance, were first proposed by *Hu (1962)*. These moments can serve as useful feature representation of images, thus allowing class recognition by suitable statistical learning (*i.e.* machine learning) algorithms. Subsequently, *Teague (1980)* showed that continuous orthogonal moments based on orthogonal polynomials such as the Legendre polynomials (see *Szegö (1975)*) and the Zernike polynomials (*von Zernike, 1934*) enable approximate image reconstruction. Examples of other continuous orthogonal moments include the pseudo-Zernike moments (*Teh & Chin, 1988*), the Gegenbauer moments (*Liao et al., 2002*), and the generalized pseudo-Zernike moments (GPZM; *Xia et al. (2007)*).

Discrete orthogonal moments based on the Chebyshev polynomials (see *Szegö (1975)*) were introduced to overcome the problem of computational complexity of continuous orthogonal moments, and they enable exact image reconstruction (*Yap, Raveendran & Ong, 2001*; *Mukundan, Ong & Lee, 2001*). An important member of this class of discrete orthogonal moments is the Krawtchouk moments (KM), which is unique for being able to extract local features in images (*Yap, Paramesran & Ong, 2003*). Other members include the Hahn moments (*Zhou et al., 2005*), dual Hahn moments (*Zhu et al., 2007*) and the Racah moments (*Zhu et al., 2007*).

In applications, orthogonal moments are widely used for non-trivial image analysis tasks, such as the identification of written alphabets in different languages (*e.g. Liao & Pawlak (1995)*, *Bailey & Srinath (1996)*). In biology, they have been used in the analysis of complex biological images, for tasks like classification of cellular subtypes (*Ryabchykov et al., 2016*), bacteria strains (*Bayraktar et al., 2006*), ophthalmic pathologies (*Adapa et al., 2020*), cancer cell phenotypes (*Alizadeh et al., 2016*), breast cancer phenotypes (*Tahmasbi, Saki & Shokouhi, 2011*; *Narváez & Romero, 2012*; *Saki et al., 2013*; *Cordeiro, Santos & Silva-Filho, 2016*), fingerprint identification (*Kaur & Pannu, 2019*), and facial recognition (*Akhmedova & Liao, 2019*).

Presently, the ease of acquiring image data from biology and medicine has created the possibility of mimicking human expert classification decisions using a purely data-driven approach *via* machine learning models. While state-of-the-art deep learning algorithms (*LeCun, Bengio & Hinton, 2015*), which use image pixel data directly from images, are currently in vogue for image-based machine learning applications, they are not suitable for initial exploratory work where data are limited. In addition, technical and infrastructural know-how to properly execute and interpret results from deep learning algorithms are

substantial barriers for the diffusion of deep learning to many areas in biology and medicine.

Might the method of orthogonal moments become increasingly redundant in biological image analysis against a background of unrelenting shift towards deep learning methods? To better understand this situation, we performed a case study to assess the usefulness of KM and GPZM as image features in classification problems involving biological images. In this paper, we address two classification problems in biology using images of varying degree of complexity. The first problem concerns fly species identification using patterns of wing venation. Specifically, we aim to contrast the quality of classifying fly species using KM features extracted from wing image data compared to using landmark data from standard geometric morphometric approach. The second problem concerns breast mass classification using information from digital mammograms. Along with several expert features, we explore how global features extracted from GPZM, and local features extracted from KM help improve classification of benign and malignant breast masses. Here, we consider variations in wing venation patterns to be relatively simple compared to variations in breast mass patterns, which are highly heterogeneous (*Aleskandarany et al., 2018*).

In the following subsections, we provide the background of the two problems.

## Problem 1: fly wing venation patterns for species identification

Generally, identifying a biological specimen down to the species level with certainty requires a certain level of taxonomic expertise. The taxonomist examines morphological characteristics of the specimen physically, and applies expert judgement to classify the specimen. This process is often slow and expensive. Additionally, taxonomists may also be increasingly hard to find in the future, as the number of permanent positions stagnate or shrink as a consequence of lack of funding and training at the tertiary level (*Britz, Hundsdörfer & Fritz, 2020*).

Traditional morphometric analysis (*Marcus, 1990*), which mainly captures size variation, or geometric morphometric analysis (*Bookstein, 1991*; *Adams, Rohlf & Slice, 2013*), which captures shape variation, are possible quantitative methods that potentially allow a data-driven approach to species identification. Landmark-based geometric morphometrics relies on using homologous landmarks, which can be unambiguously identified on an image. However, depending on the organism of interest, it is possible that few or no homologous landmarks are available, despite the fact that biological shape variation is apparent (*e.g.* cellular shape, claw shape) to the human observer.

Species identification by analysis of wing venation patterns often leads to correct identification at the species or even subpopulation level because the main source of variation in wing venation patterns is evolutionary divergence between taxa, with only rare and incomplete secondary convergence (*Perrard et al., 2014*). Currently, there is persistent interest in applying geometric morphometric analysis of wing venation patterns as a basis for identifying forensically important flies (*e.g. Sontigun et al. (2017)*, *Sontigun et al. (2019)*). Recently, *Khang et al. (2021)* provided proof-of-concept that the species identities of forensically important flies predicted using wing venation geometric

morphometric data with random forests are highly concordant with those inferred from DNA sequence data. Since analysis of whole wing image is likely to yield higher resolution data, we hypothesize that this will yield improved species prediction performance compared to using geometric morphometric landmark data which is relatively low resolution. Indeed, *Macleod, Hall & Wardhana (2018)* reported encouraging results from an image analysis of fly wing venation patterns using pixel brightness as features. However, their method requires the use of undamaged wings and a standardized protocol to minimize imaging artefacts arising from slide preparation (*e.g.* bubbles, lighting variation). Therefore, capturing information in image pixel data as translation, rotation and scale-invariant features may improve usability of images without the need to apply a rigid imaging protocol.

Here, we do not consider comparison against Elliptic Fourier Analysis (*Kuhl & Giardina, 1982*), another shape analysis method, since it is used for shapes that are closed contours, which wing venation patterns are not.

### Problem 2: breast mass classification

The classification of breast masses based solely on digital mammograms is a challenging problem, owing to the heterogeneous morphology of breast masses (*Aleskandarany et al., 2018*). Several researchers who used orthogonal moments to construct image features for the classification of benign and malignant masses reported encouraging findings (*Tahmasbi, Saki & Shokouhi, 2011*; *Narváez & Romero, 2012*). Current state-of-the-art deep learning approach to image analysis of breast cancer mammograms gave highly optimistic results. *Shen et al. (2019)* reported sensitivity of 86.7%, and specificity of 96.1% in the classification of benign and malignant breast masses, using 2478 images in the CBIS-DDSM database (training set size = 1903; validation set size = 199; test set size =376; (*Clark et al., 2013*; *Lee et al., 2017*)). Nevertheless, the opacity and plasticity of powerful black-box methods such as deep learning pose challenges to their formal adoption in medical practice (*Nicholson Price, 2018*). In the end, combining the complementary strengths of features derived from human expert judgement and those from statistical learning models seems to be the most convincing approach (*Gennatas et al., 2020*).

Here, we hypothesize that integrating the result of statistical learning outcome from image analysis using orthogonal moments with relevant expert features may ameliorate performance deficiencies based solely on image analysis.

## MATERIALS AND METHODS

### Fly wing images

For the problem of fly species identification using images of wing venation patterns, we used species from two forensically important fly families: Sarcophagidae and Calliphoridae (*Amendt et al., 2011*). Images of wings of male specimens that are of sufficiently good quality for image analysis, and their associated geometric morphometric data from 19 landmarks (Fig. 1) were taken from *Khang et al. (2020)*.

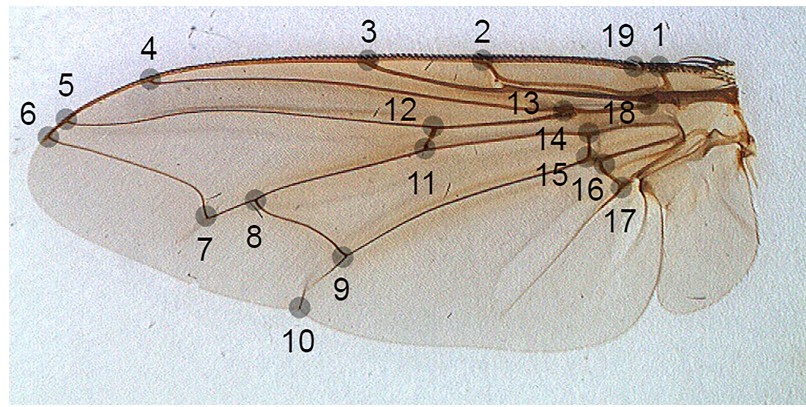

**Figure 1** **The position of landmarks (gray circles) on a sample wing image.**

The samples were taken from male flies from the Calliphoridae family (seven species), namely *Chrysomya megacephala* (Fabricius, 1794) ($n = 5$), *Chrysomya nigripes* Aubertin, 1932 ($n = 5$), *Chrysomya pinguis* (Walker, 1858) ($n = 5$), *Chrysomya ruficacies* (Macquart, 1842) ($n = 5$), *Chrysomya villeneuvi* Patton, 1922 ($n = 5$), *Lucilia cuprina* (Wiedemann, 1830) ($n = 4$) and *Lucilia porphyrina* (Walker, 1856) ($n = 5$). From the Sarcophagidae family (eight species), we had *Boettcherisca javanica* Lopes, 1961 ($n = 5$), *Boettcherisca karnyi* (Hardy, 1927) ($n = 5$), *Boettcherisca peregrina* (Robineau-Desvoidy, 1830) ($n = 5$), *Sarcophaga ruficornis* (Fabricius, 1794) ($n = 5$), *Sarcophaga dux* Thompson, 1869 ($n = 5$), *Parasarcophaga albiceps* (Meigen, 1826) ($n = 5$), *Parasarcophaga misera* (Walker, 1849) ($n = 5$) and *Sarcophaga princeps* Wiedemann, 1830 ($n=5$). In total, 74 specimens from 15 different species were used.

## Breast cancer images and associated expert features

The DDSM (Digital Database for Screening Mammography) database (*Heath et al., 1998, 2000*) is a public repository of breast cancer mammograms. Although this database contains a large collection of 2,620 scanned film mammograms, the quality of annotations in the images varies. Examples of errors include wrongly annotated images and lesion outlines that do not form precise mass boundary (*Lee et al., 2017*; *Song et al., 2009*). Inclusion of such poor quality images in the training phase of a statistical learning model can weaken model generalizability.

To overcome this problem, a curated subset of images in the DDSM database, known as the CBIS-DDSM (Curated Breast Imaging subset of DDSM) collection (*Clark et al., 2013*; *Lee et al., 2017*) was created. Images in this database consist of selected mammograms that have been segmented using an automated segmentation algorithm. The segmented images were evaluated by comparing outlines of mass lesion images with hand-drawn outlines made by a trained radiologist. The CBIS-DDSM collection comprises scanned filmed mammography from 1,566 participants. A patient could have more than one type of lesions (*e.g.* mass, calcification) in a single mammogram. The images were decompressed and converted to the DICOM format containing updated region of interest (ROI)

segmentation, bounding boxes, and pathologic diagnosis. Moreover, a total of 3,568 focused images are also available in this database to cater to studies that do not require the use of full mammogram images but only a focused region of abnormalities. We used 1,151 images from the CBIS-DDSM collection, of which 48% (556/1,151) are of benign class, and 52% (595/1,151) are of malignant class. The images were downloaded from the CBIS-DDSM database in the PNG file format.

Each mammogram image is further associated with five expert features: (i) BI-RADS assessment; (ii) mass shape; (iii) mass margin; (iv) breast density; (v) subtlety rating. The Breast Imaging-Reporting and Data System (BI-RADS) provides a standard for reporting breast examination results based on mammography, ultrasound, and magnetic resonance imaging data. First published in 1992 (*American College of Radiology, 1992*), BI-RADS has become a standard communication tool for mammography reports globally (*Balleyguier et al., 2007*), and is now in its fifth edition (*Sickles et al., 2013*). By standardizing the reporting of mammography results, BI-RADS facilitates communication among radiologists and clinicians, and aids the training and education of junior radiologists in developing countries (*Lehman et al., 2001*).

There are seven categories in the BI-RADS assessment that can be assigned based on evaluation of the lexicon descriptors or the biopsy findings of a lesion (*Sickles et al., 2013*). Category 0 indicates that materials are insufficient for evaluation and additional imaging evaluation or prior mammograms for comparison are required. Category 1 is given when no anomalies are found. Category 2 is given when there is evidence of benign tumors such as skin calcifications, metallic foreign bodies, fat-containing lesions and involuting calcified fibroadenomas. If radiologists are unsure of the lesion categorization, a BI-RADS category of 3 is given, and a follow-up over a certain interval of time is done to determine stability of the lesion. The risk of malignancy in this category is considered to be at most 2%. Category 4 is assigned when malignant tumors are suspected. Three subcategories are possible: a, b, and c. The subcategory (a) reflects a subjective probability of malignancy between 2% to 10%. Subcategory (b) reflects a subjective probability of 10% to 50% for malignancy, while subcategory (c) reflects a subjective probability of malignancy ranging from 50% to 95%. Category 5 reflects a strong belief (probability of 95% or more) that a lesion is malignant. When a lesion placed in this category is contradicted by a benign biopsy report, a surgical consultation may still be advised. Finally, a BI-RADS category of 6 is given when a lesion receives confirmation of malignancy from the biopsy result.

A breast mass has two important aspects: shape and margin. BI-RADS lexical descriptors of mass shape include oval, irregular, lobulated, etc. The mass margin describes the shape of the edges of a mass, such as being circumscribed, ill-defined, or spiculated. Breast density (*Sickles et al., 2013*) is a categorical variable with four categories. Category 1 describes breast composition that is almost entirely fatty. Category 2 indicates the presence of scattered fibroglandular densities. Category 3 indicates breast that is heterogeneously dense. Category 4 indicates breast that is extremely dense, which lowers sensitivity of mammography. Finally, the subtlety rating, which is not part of the BI-RADS standards, is an ordinal variable on a scale of 1 to 5 representing the difficulty in viewing the

abnormality in a mammogram (*Lee et al., 2017*). The scale ranges from 1 for "subtle" to 5 for "obvious".

## Data processing

### *Fly wing images*

For the fly images, the images could not be used directly because of the presence of non-biological variation such as damaged wing membranes, lighting variation, and presence of bubbles in the slides. We processed the images by converting them into binary images using Pinetools (https://pinetools.com/threshold-image), with a focus on retaining the pattern of venation on the wing. Subsequently, we denoised the binary images manually. All images were cropped to a uniform size of $724 \times 254$ pixels. The final images were stored in PNG format.

### *Breast cancer images*

The regions of interest associated with the CBIS-DDSM breast cancer images were resized to a uniform size of $300 \times 300$ pixels, and normalized using the EBImage R package (Version 3.0.3; *Pau et al. (2010)*). Subsequently, we enhanced the contrast in the ROI of images using the histogram equalization method (*Gonzalez & Woods, 2002*).

## Feature extraction

### *Fly wing images*

To use geometric morphometric data from the wing images, raw coordinate data from the 19 landmarks were processed using Generalized Procrustes Analysis in the geomorph R package (Version 3.3.2; *Adams & Otarola-Castillo, 2013*) to produce the translation, rotation, and scale-invariant Procrustes coordinates. To remove the effect of allometry, we used the residuals produced from linear regression of the Procrustes coordinates against the logarithm (base 10) of centroid size (*Sidlauskas, Mol & Vari, 2011*; *Klingenberg, 2016*). To remove correlation between the Procrustes coordinates, we applied R-mode principal component analysis (PCA), and kept the first 15 principal components that cumulatively explain 98.7% of the total variation in the data.

For image analysis, since the fly wing images are rectangular, we used higher order moments to ensure that the reconstruction captured image details distal from the image centroid (Fig. 2). We found that Krawtchouk moment invariants (see Appendix) of order 200 was appropriate for extracting image features from the binary images of the fly wings. Thus, 40,000 moment invariant features were obtained. Since the range of values for these features was generally large, we scaled these features using the Z-score, and then applied Q-mode PCA to reduce the dimension of the feature space. The first 60 principal components accounting for 92.6% of total variance were used as features for downstream statistical learning work.

### *Breast cancer images*

For the breast cancer images, we used KM of order 163. For GPZM, we used order 126, and set $\alpha = 0$. The choice of $\alpha$ value was guided by results in *Xia et al. (2007)*, which showed

**(A)**

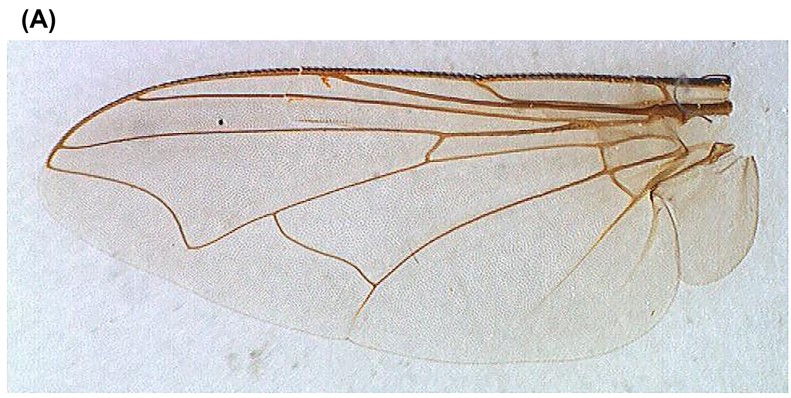

**(B)**

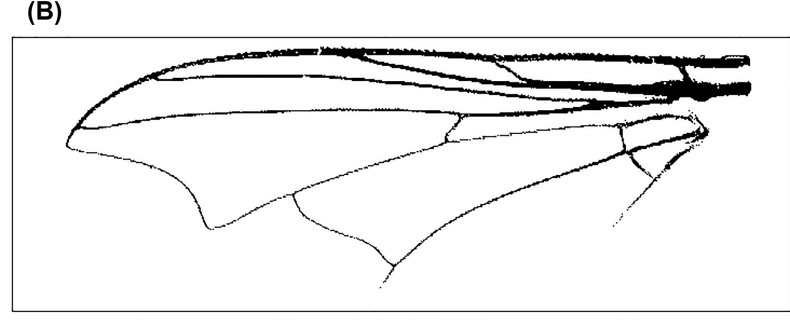

**(C)**

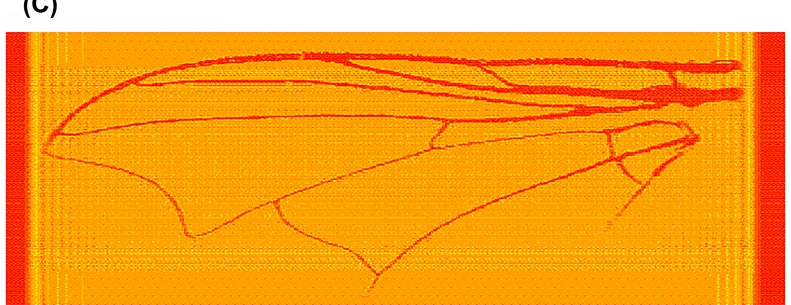

**Figure 2  An example of fly wing image data taken from *C. nigripes*.** (A) Raw image; (B) binary image after manual denoising; (C) reconstructed image using Krawtchouk moment invariants of order 200.

that reconstruction error tended to be relatively smaller for $\alpha = 0$ compared to larger values of $\alpha$, under the assumption of the presence of low level noise in the images.

The KM and GPZM orders were estimated using data as follows. First, we randomly picked 8 images from the CBIS-DDSM collection and resized the images to a uniform size of 300 x 300 pixels. The images were then normalized and enhanced using the histogram equalization method. Subsequently, the GPZM and KM moments from order 1 to order 300 were computed for the images. For each order, we reconstructed the images and calculated the mean squared error (MSE) for each image. The MSE can be obtained by squaring the difference of the reconstructed image from the original image followed by averaging using the dimension of the image.

For KM, order 163 was used as it produced the lowest MSE error, with mean and standard deviation of the reconstructed images being 0.0128 and 0.0030, respectively. For GPZM, the lowest MSE ranged from order 11 to order 238 with mean of 0.0028 and standard deviation of 0.0044, respectively. Order 126 was the mean of the orders producing the lowest MSE among the 8 images. The mean and the standard deviation of the reconstructed images with order 126 were 0.0040 and 0.0060, respectively. Plots of the MSE for orders 1 to 300 for each of the 8 images are given in Supplemental File 1.

The top 1% of features with the largest magnitude of t-statistic values were selected as the feature vector. We then applied Q-mode PCA on these selected features, and used the first $k$ principal components that explain about 95% of the total variance.

## Study design and analysis

### Data analysis

For Problem 2, consider an $n \times p$ input data matrix that has been subjected to Q-mode PCA. This produces $W^T$, the $p \times n$ matrix of principal component loadings. With $p > n$, as is the case with features extracted using orthogonal moments, the matrix of principal component scores $V$ is of dimension $n \times n$. A partial principal component scores matrix $V_\beta$ accounting for $\beta$ proportion of total variation in the training samples is the $n \times k$ submatrix obtained from $V$ by taking the first $k$ columns of $V$. Applying Fisher's linear discriminant analysis using $V_\beta$, we then obtain the $k \times (s - 1)$ matrix of weights $A$ for the $s-1$ linear discriminants, where $s$ is the number of classes.

Given an $n_{\text{test}} \times p$ matrix of test samples $X_{\text{test}}$, we first center the test samples $X_{\text{test, centered}} = X_{\text{test}} - (\mu_{\text{train}},\ldots,\mu_{\text{train}})^T$, where $\mu_{\text{train}}^T$ is the $1 \times p$ vector of mean of each of the $p$ variables in the training set. Then, we map the test samples into the principal component space of training samples using the matrix operation $V_{\text{test}} = X_{\text{test, centered}} W^T$. Thereafter, we obtain the partial principal component scores matrix $V_{\text{test}, \beta}$, which is of dimension $n_{\text{test}} \times k$, and finally map the test samples into the linear discriminant space of the training samples using the matrix product $V_{\text{test}, \beta} A^T$., which is of dimension $n_{\text{test}} \times (s-1)$. We use the latter for testing.

### Statistical learning model

For classification, we used a kernel discriminant model in the GUIDE (Generalized, Unbiased, Interaction Detection and Estimation) classification and regression tree program (Loh, 2009, 2014). The kernel method is a non-parametric method that estimates a Gaussian kernel density (Silverman, 1986) for each class in a node, and uses the estimated densities in a maximum likelihood framework for classification. The tree complexity parameter $k$-SE in GUIDE was set at the default value of 0.5, with the number of cross-validated trees set at 10.

For Problem 1, it was not feasible to split the samples into another test set for assessing generalization error, because the average sample size per class was already small (about 5). Hence, we applied the random forests ensemble classifier (2001 trees) and obtained the out-of-bag error estimate for generalization error.

For the CBIS-DDSM data set in Problem 2, we randomly chose 70% of the data set for training, and the remainder 30% for testing. A total of 10 such instances were made to study variation in performance metrics. Subsequently, the predicted probability of malignancy (P(mal)) was used as a feature together with subsets of the expert features to build another machine learning model. For this, we considered four models: (I) expert features with BI-RADS assessment and P(mal) from image analysis; (II) expert features with BI-RADS assessment, without P(mal); (III) expert features with BI-RADS assessment replaced by P(mal) from image analysis; (IV) expert features without BI-RADS assessment. To understand the relative contribution of the set of variables in the four models, we computed variable importance scores for each variable in Models I to IV using GUIDE, following the method in *Loh (2012)*. We ranked the importance scores of each variable in the four models (1 being most important) and then reported the mean and standard deviation of the ranks.

## Performance evaluation

We used standard metrics for the evaluation of classifier performance. Accuracy is defined as the probability that the predicted class is the same as the true class. For multi-class prediction in the fly images, we did not use sensitivity or specificity, as contextually no particular species is of any special interest. Hence, sensitivity and specificity were considered only in the breast cancer classification. There, sensitivity is defined as the probability of predicting the malignant class, given that a sample is malignant. Specificity is defined as the probability of predicting the benign class, given that a sample is benign.

In the case of the fly images, we used the Bayesian posterior mean of accuracy with uniform prior (see Appendix), and reported the 95% Bayesian credible interval (*Brown, Cai & DasGupta, 2001*). For the breast cancer images, we reported the mean of accuracy, sensitivity, and specificity from the 10 random instances, along with their associated standard error estimate (sample standard deviation $/\sqrt{10}$).

## Software and computation

For image analysis, we used R version 3.6.1 (*R Core Team, 2018*) to perform the computations and run the IM R package (*Rajwa et al., 2013*). Data processing and analyses of the CBIS-DDSM samples and fly wing images were done using a 22 CPU core, 23 GB RAM server running on Ubuntu 16.04.4 LTS at the Data Intensive Computing Centre, Universiti Malaya, Malaysia. For classification using decision trees with kernel model and random forests, we used the GUIDE program (Version 35.2; *Loh (2009*, *2014)*).

# RESULTS

## Quality of moment feature representation of images

For the fly wings, images reconstructed from KM of order 200 approximated the binary images very well (Fig. 2C). Similarly, images reconstructed from KM (order 163) and GPZM ($\alpha = 0$) for the breast images also approximated the ROI well.

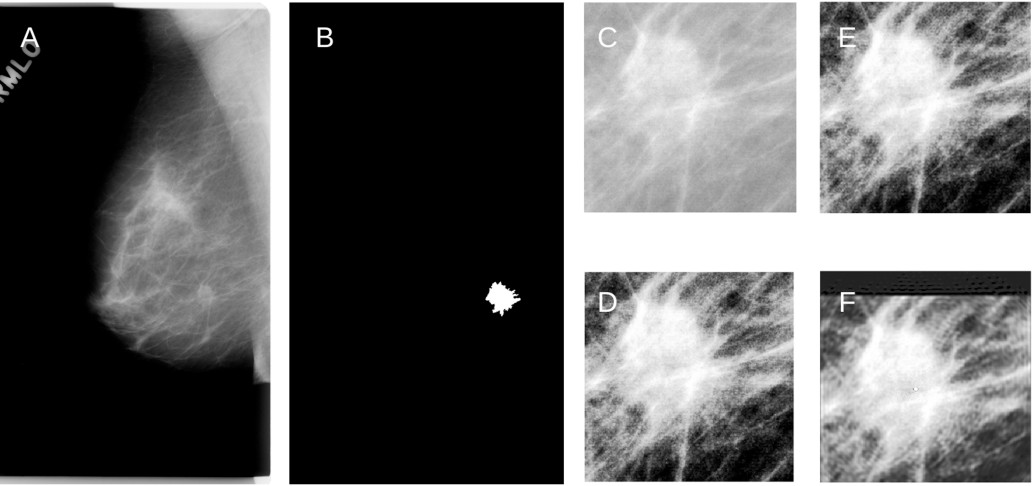

**Figure 3 (A) An example of the raw mammogram image showing malignant spiculated breast mass (index no. P_01461).** (B) Image after thresholding showing the tumor (white). (C) Region of interest centered on the mass (390 × 385 pixels) before (D) Enhancement. (E) Reconstruction using KM; (F) Reconstruction using GPZM.

An example showing a malignant mass is given in Fig. 3. For additional examples, see Supplemental File 2.

## Classification of fly species

Krawtchouk moment invariants of order 200 captured important patterns of variation in wing venation that led to 94.7% correct classification of 74 training samples (95% Bayesian credible interval for accuracy = [88.8–98.5%]). Indeed, within species variation was substantially smaller for 12 species, compared to between species variation (Fig. 4). In comparison, the use of geometric morphometric data produced classification accuracy of 64.5% (95% Bayesian credible interval for accuracy = [53.5–74.8%]). Baseline prediction accuracy using the majority class was 7.9%. The classification matrices for both cases are given in Supplemental File 3.

Under the random forests model, predicting species using Krawtchouk moment invariants produced 100% accurate classification result, whereas two errors were made when geometric morphometric data was used. Out-of-bag error using Krawtchouk moment invariants was 0% (0/74), compared to 39.2% using geometric morphometric data (29/74).

## Classification of benign and malignant breast masses

The mean classification accuracy based solely on image data was about 57% ± 1%, with mean sensitivity about 70% ± 1%, and mean specificity of 43% ± 2%. Baseline prediction accuracy using majority class was 52%. Figure 5 shows, for a particular training-testing instance, the estimated bivariate Gaussian kernel densities in the space of the first linear discriminants derived from KM and GPZM, with test samples superimposed on the plot.

When expert features were used together with P(mal) as a feature, we observed substantial increase of mean accuracy (from 57% to between 68% and 75%; Table 1), mean

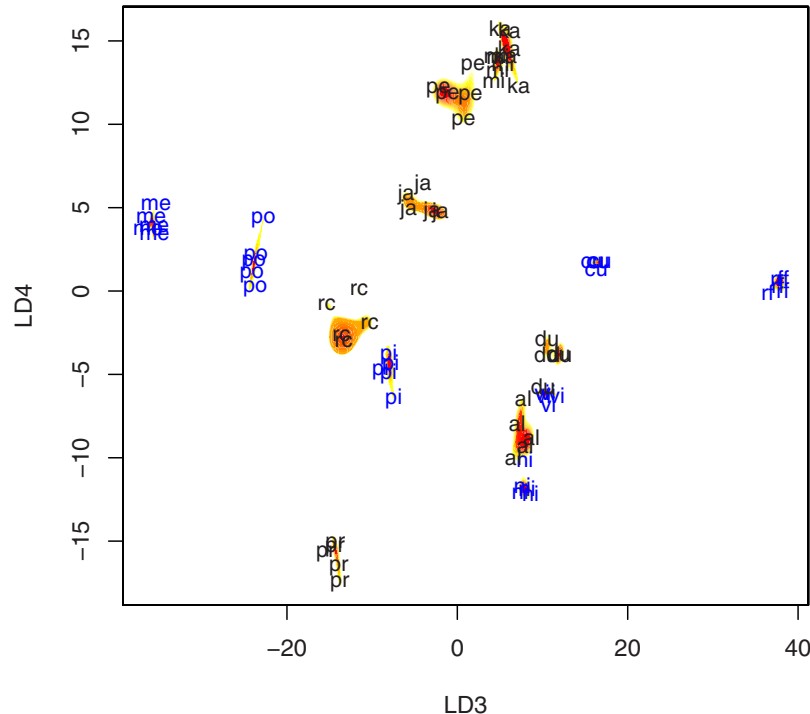

**Figure 4 Scatter plot of fly wing shape in LD3 and LD4 space, with contour plots of the estimated species-specific bivariate Gaussian kernel densities (heat-colored regions).** Abbreviations: al, *P. albi-ceps*; cu, *L. cuprina*; du, *S. dux*; ja, *B. javanica*; ka, *B. karnyi*; me, *C. megacephala*; mi, *P. misera*; ni, *C. nigripes*; po, *L. porphyrina*; pi, *C. pinguis*; rc, *S. ruficornis*; rf, *C. rufifacies*; pe, *B. peregrina*; pr, *S. princeps*; vi, *C. villeneuvi*. LD3, third linear discriminant score; LD4, fourth linear discriminant score. Color annotation: black for Sarcophagidae species; blue for Calliphoridae species.

sensitivity (from 70% to between 80% and 97%), and mean specificity (from 43% to between 51% and 61%, excluding Model IV). Using the set of expert features without P(mal) (Model II) produced the best mean accuracy (75% ± 1%) and best mean sensitivity (97% ± 0%).

Removal of the BI-RADS assessment (Model IV) affected mean specificity substantially, causing it to drop from 51% (Model II) to 40%. If BI-RADS assessment was replaced by P(mal), mean specificity improved slightly, at the expense of mean sensitivity, which dropped from 97% (Model II) to 80%, thus causing a concurrent drop in mean accuracy from 75% to 68%. This suggests that the information in BI-RADS assessment and P(mal) are non-redundant and may complement each other. Finally, these two features interact to produce a model (Model I) where the difference in sensitivity and specificity is minimized (24%) compared to other models (Model II: 46%; Model III: 34%; Model IV: 55%). Thus, the inclusion of P(mal) as a feature seemed to be important for reducing the discrepancy between model sensitivity and model specificity by decreasing the former but increasing the latter.

Table 2 shows summary statistics of feature importance (mean ± standard deviation) for each of the four models. Across all four models, BI-RADS assessment (where used) and mass margin were consistently ranked as the two most important features, and the order of

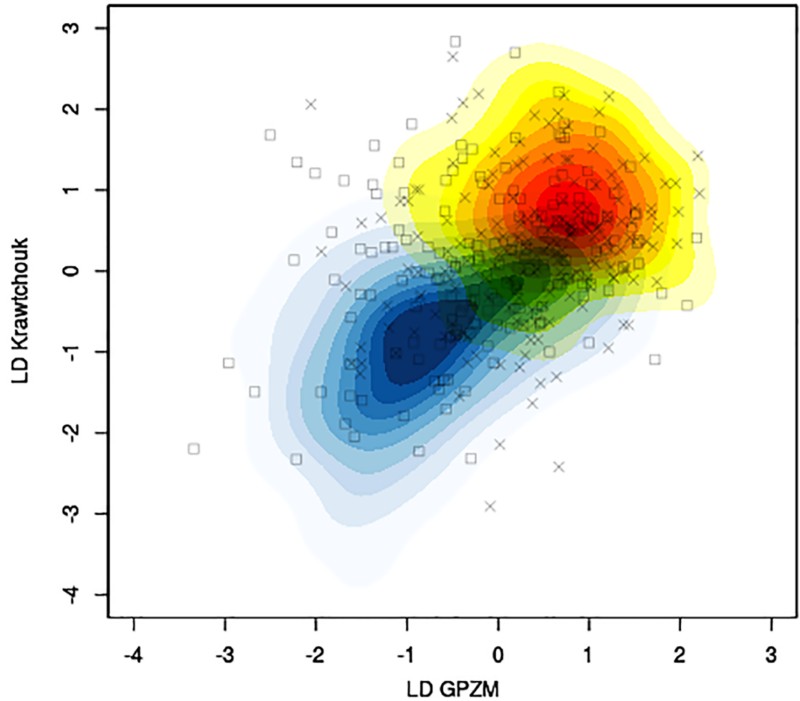

**Figure 5 Example of contour plots of estimated bivariate Gaussian kernel densities for benign (light blue to dark blue tones) and malignant (yellow to red tones) training data (CBIS-DDSM data with seed 261) in the space of linear discriminants (first) based on Krawtchouk moments and generalized pseudo-Zernike moments.** Squares and crosses indicate benign and malignant test samples, respectively.

**Table 1 Mean accuracy, sensitivity and specificity (±standard error) of Models I to IV, in percentages.**

| Model | Accuracy | Sensitivity | Specificity |
|---|---|---|---|
| I | 70 ± 1 | 85 ± 1 | 61 ± 2 |
| II | 75 ± 1 | 97 ± 0 | 51 ± 1 |
| III | 68 ± 2 | 80 ± 1 | 54 ± 3 |
| IV | 69 ± 1 | 95 ± 1 | 40 ± 3 |

**Table 2 Variable (feature) importance of the predicted probability of malignancy (P(mal)) and the five expert features for Models I to IV. The most important variable is ranked 1, with larger ranks indicating less importance. Abbreviation: NA, not available.**

| Model | P(mal) | Assessment | Mass margins | Mass shape | Subtlety | Breast density |
|---|---|---|---|---|---|---|
| I | 2.8 ± 0.6 | 1.5 ± 0.5 | 1.7 ± 0.6 | 4.0 ± 0.0 | 5.0 ± 0.0 | 6.0 ± 0.0 |
| II | NA | 1.6 ± 0.5 | 1.4 ± 0.5 | 3.0 ± 0.0 | 4.0 ± 0.0 | 5.0 ± 0.0 |
| III | 1.9 ± 0.3 | NA | 1.1 ± 0.3 | 3.0 ± 0.0 | 4.0 ± 0.0 | 5.0 ± 0.0 |
| IV | NA | NA | 1.0 ± 0.0 | 2.0 ± 0.0 | 3.0 ± 0.0 | 4.0 ± 0.0 |

importance (in decreasing importance) for mass shape, subtlety, and breast density was also consistent. Where P(mal) was used in Model I, on average it ranked third in order of importance; in Model III, on average it ranked second, after mass margin. This suggests that P(mal), which summarizes abstract information from image data, provides useful information to the statistical learning model when used together with the expert features.

## DISCUSSION

Currently, geometric morphometrics is the standard method for the analysis of wing shape variation in entomology (*Tatsuta, Takahashi & Sakamaki, 2018*), of which species identification is only one possible application. Nevertheless, for routine species identification, direct image analysis can also be practical and produce more accurate prediction results. Indeed, we showed that images containing artefacts could still be useful for species identification, as such artefacts can be removed in the binary images *via* manual denoising. Krawtchouk moment invariants of images can be generated quickly, thus making them useful as sample features for statistical learning models in the initial evaluation of the difficulty of a biological image classification problem. Indeed, in situations where only a few landmarks can be reliably identified, image analysis using orthogonal moments is a feasible alternative to addressing the species identification problem.

Given the encouraging results of applying Krawtchouk moment invariants to image analysis of fly wing venation patterns in the present study, we conjecture that image-based identification of other insects where substantial species-specific variation is present in the wing organs, such as dragonflies (*Kiyoshi & Hikida, 2012*) and mosquitoes (*Lorenz et al., 2017*), may also be fruitful.

For classification of breast masses using KM and GPZM, the result was less satisfactory. A potential source of error may be noise generated in some ROI images that were originally smaller (*e.g.* $159 \times 95$ pixels) when they were rescaled to $300 \times 300$ pixels. Several studies that used orthogonal moments in breast mass classification reported apparently optimistic results, but their study design should be considered carefully. For example, *Tahmasbi, Saki & Shokouhi (2011)* reported classification accuracy of 96%, sensitivity of 100%, and specificity of 95%. They used a different set of breast cancer images ($n = 121$) from the much smaller Mini-MIAS database (*Suckling et al., 1994*), of which about 55% ($n = 67$) are benign cases, and 45% ($n = 54$) are malignant cases. As a result, the number of images available for testing ($n \approx 36$, with benign and malignant cases being approximately equal) was limited. By introducing manual segmentation in their work to accentuate mass boundaries in the ROIs, the authors inadvertently injected expert knowledge into the study. They extracted features using Zernike moments (*Khotanzad & Hong, 1990*) and applied artificial neural network as the classifier. *Narváez & Romero (2012)* used KM and Zernike moments to extract features from images in the DDSM database for the classification of breast masses. They reported test accuracy of about 90% with KM features. However, the size of the test samples was small ($n = 100$; half benign, half malignant). It was also unclear from their study design whether the selected training samples ($n = 300$) and test samples were randomized.

The United States of America national performance benchmark on screening mammography by radiologists recently established a mean sensitivity 86.9% (95% confidence interval [86.3–87.6%]) and a mean specificity of 88.9% (95% confidence interval [88.8–88.9%]), using a sample of 359 radiologists who examined about 1.7 million digital mammograms (*Lehman et al., 2017*). In comparison, deep learning algorithms produced impressive prediction performance that was seemingly on par with, or surpassed expert performance. Using 2,478 images in the CBIS-DDSM database (training set size = 1,903; validation set size = 199; test set size = 376), *Shen et al. (2019)* reported sensitivity of 86.7%, and specificity of 96.1% in the classification of malignant and benign masses. However, the apparent optimism in deep learning as the last word in medical image analysis was recently questioned. *Wang et al. (2020)* reported that deep learning test accuracy dropped substantially when test samples with distribution of patterns of variation that differed from that of the validation samples' were used to challenge the trained deep learning model. It seems that allowing image information to interact with expert features may produce more robust models when we attempt to classify biologically complicated images such as mammograms.

## CONCLUSIONS

Through the analysis of fly wing images, we showed that orthogonal moments such as the Krawtchouk moments are effective features for summarizing meaningful patterns in relatively simple biological images. The GUIDE kernel model that uses Krawtchouk moment invariants gave highly accurate prediction of all 15 fly species studied, beating the similar model that uses geometric morphometric data by a wide margin.

On the other hand, the efficacy of orthogonal moments-based features for summarizing patterns of variation that are more heterogeneous and less well-defined in complex biological images appears modest. Through analysis of the CBIS-DDSM breast mammograms, we found statistical learning models that use orthogonal moments produced classification performance that was far below those achieved by trained radiologists. Nevertheless, when output of the predictive model in the form of predicted probability of malignancy was used as a feature to summarize image evidence for the malignancy class, we found its variable importance score surpassed those of expert features associated with mammograms (*e.g.* mass shape, breast density, and subtlety rating). We also found the predicted probability of malignancy to interact with the important BI-RADS assessment for malignancy expert feature, leading to prediction performance that is optimal in the sense of having the smallest discrepancy between sensitivity and specificity.

To summarize, we believe orthogonal moments are still feasible as image features in the analysis of biological images. They should be adequate for handling species prediction problems on the basis of the shape of specific anatomies. The ease of applying them means that orthgonal moments are ideal for estimating a lower bound of prediction performance. On the other hand, expert features that accompany more complex biological images are probably necessary to offset the modest performance of statistical learning models that use orthogonal moments for class prediction.

## APPENDIX

### Orthogonal moments

In this section, we provide sufficient mathematical background for the appreciation of the use of orthogonal moments as feature extractors of images. For further advanced details, we refer readers to *Szegö (1975)* for the theory of orthgonal polynomials, *Yap, Paramesran & Ong (2003)* for Krawtchouk moments, *Xia et al. (2007)* for generalized pseudo-Zernike moments, and *Shu, Luo & Coatrieux (2007)* for a general introduction to using orthogonal moments for image analysis.

### Mathematical preliminaries

**Definition 1.** (*Szegö, 1975*) *Let $p_n(x)$ be a polynomial in x of order n. For the interval $a \leq x \leq b$, if $w(x)$ is a weight function in x, and $\delta_{nm}$ is the Kronecker delta which is equal to 1 when $n = m$, and 0 when $n \neq m$, then $p_n(x)$ is said to be an orthogonal polynomial associated with the weight function $w(x)$ if it satisfies the condition*

$$\int_a^b p_n(x)p_m(x)w(x)dx = \delta_{nm}.$$

**Definition 2.** (*Oberhettinger, 1964*) *The hypergeometric function, ${}_2F_1(a,b;c;z)$ is a special function defined as the power series*

$${}_2F_1(a, b; c; z) = \sum_{k=0}^{\infty} \frac{(a)_k (b)_k}{(c)_k} \frac{z^k}{k!},$$

where *a,b,c* are real numbers, with $|z| < 1$. The notation $(a)_k$ denotes the Pochhammer symbol for the rising factorial $(a)_k = a(a + 1)(a + 2)\ldots(a + k − 1)$, with $(a)_0 = 1$.

### Krawtchouk polynomials and moments

The Krawtchouk polynomials (*Krawtchouk, 1929a*, *1929b*) are discrete orthogonal polynomials associated with a binomial probability weight function. The Krawtchouk polynomial of order *n* is denoted by $k_n(x; p, N − 1)$, and can be conveniently representing as a hypergeometric function

$$k_n(x; p, N − 1) = {}_2F_1\left(-n, -x; -N + 1; \frac{1}{p}\right)$$

where $n, x = 0, 1, 2, \ldots, N − 1, N > 1, 0 < p < 1$. The weighted Krawtchouk polynomial (*Yap, Paramesran & Ong, 2003*) $\overline{k}_n(x; p, N − 1)$ is given by

$$\overline{k}_n(x; p, N − 1) = k_n(x; p, N − 1)\sqrt{\frac{\omega(x; p, N − 1)}{\rho(x; p, N − 1)}},$$

where $\omega(x; p, N − 1)$ is the binomial probability mass function

$$\omega(x; p, N − 1) = \binom{N − 1}{x} p^x(1 − p)^{N−1−x},$$

and $(1-p)^{N-1}/\rho(x;p,N-1) = \omega(n;p,N-1)$. For brevity, we will write $\bar{k}_n(x;p,N-1)$ as $\bar{k}_n(x)$.

Let the Krawtchouk moments matrix of order $l$ be an $l \times l$ square matrix, $\mathbf{Q}$. The $(n,m)$ element of $\mathbf{Q}$, denoted as $Q_{nm}$, is related to the image intensity function $f(x,y)$ on the two-dimensional discrete domain through

$$Q_{nm} = \sum_{x=0}^{N-1}\sum_{y=0}^{M-1}\bar{k}_n(x)\bar{k}_m(y)f(x,y).$$

Consider an image of dimension $N \times M$. If we denote $\mathbf{K}_1$ and $\mathbf{K}_2$ as the $l \times N$ and $l \times M$ matrix of weighted Krawtchouk polynomials, respectively, with $\mathbf{A}$ as the $N \times M$ matrix of image intensity functions, then

$$\mathbf{Q} = \mathbf{K_1}\mathbf{A}\mathbf{K_2^T}.$$

The orthogonality property of the weighted Krawtchouk polynomials

$$\sum_{x=0}^{N-1}\bar{k}_n(x)\bar{k}_m(x) = \delta_{nm},$$

implies that the product of an $l \times N$ matrix of weighted Krawtchouk polynomials with its transpose is the $l \times l$ identity matrix. Therefore, the image intensity functions can be reconstructed from $\mathbf{Q}$ as

$$\mathbf{A} = \mathbf{K_1}^T\mathbf{Q}\mathbf{K_2},$$

that is,

$$f(x,y) = \sum_{n=0}^{N-1}\sum_{m=0}^{M-1}\bar{k}_n(x)\bar{k}_m(y)Q_{nm}.$$

### Generalized pseudo-Zernike polynomials and moments

The pseudo-Zernike polynomials (*Bhatia & Wolf, 1954*) are polynomials in two variables that form a complete orthogonal set for the interior of the unit circle. The pseudo-Zernike polynomials of order $n$ and repetition $m$ is denoted by $V_{nm}(r,\theta)$, and expressed in polar coordinate form as

$$V_{nm}(r,\theta) = R_{nm}(r)e^{im\theta},$$

where $i$ is the complex number $\sqrt{-1}$, and $R_{nm}(r)$ is the radial polynomial defined as

$$R_{nm}(r) = \sum_{j=0}^{n-|m|}\frac{(-1)^j(2n+1-j)!}{j!(n-|m|-j)!(n+|m|+1-j)!}r^{n-j},$$

where $n = 0, 1, 2, , \infty$, and $|m| \le n$. *Xia et al. (2007)* proposed a generalization of $R_{nm}(r)$

$$R_{nm}^{\alpha}(r) = \frac{(n+|m|+1)!}{(\alpha+1)_{n+|m|+1}} \sum_{j=0}^{n-|m|} (-1)^j \frac{(\alpha+1)_{2n+1-j}}{j!(n-|m|-j)!(n+|m|+1-j)!} r^{n-j},$$

where $\alpha > -1$, with $R_{nm}^0(r) = R_{nm}(r)$. The weighted generalized radial polynomial is given by

$$\bar{R}_{nm}^{\alpha}(r) = R_{nm}^{\alpha}(r) \sqrt{\frac{(2n+\alpha+2)(\alpha+1+n-|m|)_{2|m|+1}}{2\pi(n-|m|+1)_{2|m|+1}}} (1-r)^{\alpha/2},$$

leading to the generalized pseudo-Zernike polynomials

$$\bar{V}_{nm}^{\alpha}(r,\theta) = \bar{R}_{nm}^{\alpha}(r)e^{im\theta}.$$

The generalized pseudo-Zernike moments (GPZM) of order $n$ and repetition $m$ are defined as

$$\bar{Z}_{nm}^{\alpha} = \int_0^{2\pi} \int_0^1 [\bar{V}_{nm}^{\alpha}(r,\theta)]^* f(r,\theta) r \, dr \, d\theta,$$

where $*$ denotes complex conjugate. The orthogonality property of the pseudo-Zernike polynomials implies that

$$\int_0^{2\pi} \int_0^1 \bar{V}_{nm}^{\alpha}(r,\theta)[\bar{V}_{kl}^{\alpha}(r,\theta)]^* r \, dr \, d\theta = \delta_{nk}\delta_{ml}.$$

The image can be approximately reconstructed using the inverse transform formula (*Teague, 1980*)

$$f(r,\theta) \approx \sum_{n=0}^{\infty} \sum_{|m|\leq n} \bar{Z}_{nm}^{\alpha} \bar{V}_{nm}^{\alpha}(r,\theta).$$

## Bayesian estimator of classification accuracy

Let $X_i$, $i = 1,2,\ldots,s$ be the number of correctly predicted samples for the $i$th species (up to $s$ species). These are the diagonal entries of the $s \times s$ classification matrix. For a statistical learning model, assume that it has constant probability $\pi$ of correctly predicting the species identity of a sample (*i.e.* accuracy). Then, $X_i$ is binomially distributed with number of trials equal to the number of $i$-th species in the sample ($n_i$) and success probability $\pi$. By Bayes' Theorem, the posterior distribution of $\pi$, given $X_1, X_2, \ldots, X_s$, is

$$f(\pi|X_1, X_2, \ldots, X_s) = P(X_1, X_2, \ldots, X_s|\pi)f(\pi) \times \left( \int_0^1 P(X_1, X_2, \ldots, X_s|\pi)f(\pi)d\pi \right)^{-1},$$

where $f(\pi)$ is the prior distribution of $\pi$. Using the conservative uniform prior $f(\pi) = 1$, $0 < \pi < 1$, and assuming that $P(X_1, X_2, \ldots, X_s|\pi) = \prod_{i=1}^s P(X_i|\pi)$, it can be shown that $f(\pi|X_1, X_2,\ldots,X_s)$ has a beta distribution with shape parameters $\alpha = \sum_{i=1}^s X_i + 1$, and

$\beta = N - \sum_{i=1}^{s} X_i + 1$, where $N$ is the total sample size. Thus, the Bayesian posterior mean estimate of $\pi$ is given by

$$\hat{\pi} = \frac{\sum_{i=1}^{s} X_i + 1}{N + 2}.$$

The lower and the upper limit of the 95% Bayesian credible interval of $\pi$ are computed as the 2.5th and the 97.5th percentile of the beta distribution with shape parameters $\alpha = \sum_{i=1}^{s} X_i + 1$ and $\beta = N - \sum_{i=1}^{s} X_i + 1$, respectively.

## ACKNOWLEDGEMENTS

We thank Dr. C.S. Liew and K.G. Ng from the Data Intensive Computing Centre, Universiti Malaya for technical support. We also thank Dr. Gianluca Polgar for help with graphics.

### Funding

Jia Yin Goh was supported by a research assistantship through the RU Grant from the Faculty of Science, Universiti Malaya, Malaysia (Grant number: GPF029B-2018). The funders had no role in study design, data collection and analysis, decision to publish, or preparation of the manuscript.

### Grant Disclosures

The following grant information was disclosed by the authors:
Faculty of Science, Universiti Malaya, Malaysia: GPF029B-2018.

### Competing Interests

The authors declare that they have no competing interests.

### Author Contributions

- Jia Yin Goh performed the experiments, analyzed the data, performed the computation work, prepared figures and/or tables, authored or reviewed drafts of the paper, and approved the final draft.
- Tsung Fei Khang conceived and designed the experiments, performed the experiments, analyzed the data, performed the computation work, prepared figures and/or tables, authored or reviewed drafts of the paper, and approved the final draft.

### Data Availability

The codes and data are available in the Supplemental Files. Links to external data sources are provided in README files therein.
Data availability for Problem 2:
The dataset and its metadata are available at the CBIS-DDSM website: https://wiki.cancerimagingarchive.net/display/Public/CBIS-DDSM.

The metadata of the dataset are available by following the sequence below:

CBIS-DDSM website > Data Access > Mass-Training-Description (csv) > Download

CBIS-DDSM website > Data Access > Mass- Test-Description (csv) > Download

The breast cancer images are available by following the sequence below:

CBIS-DDSM website > Detailed Description > Mass-Training ROI and Cropped Images (DICOM) > Download

CBIS-DDSM website > Detailed Description > Mass-Test ROI and Cropped Images (DICOM) > Download

For the breast cancer images, clicking the "Download" button will save a ".tcia" manifest file that can only be opened with the NBIA Data Retriever.

To download the NBIA Data Retriever, please follow the instructions in this link: https://wiki.cancerimagingarchive.net/display/NBIA/Downloading+TCIA+Images

## Supplemental Information

Supplemental information for this article can be found online at http://dx.doi.org/10.7717/peerj-cs.698#supplemental-information.

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
