# Peer review of "On the classification of simple and complex biological images using Krawtchouk moments and Generalized pseudo-Zernike moments: a case study with fly wing images and breast cancer mammograms"

_PeerJ Computer Science, doi:10.7717/peerj-cs.698_

## Round 0.1 · original submission · Major Revisions

The paper needs thorough language improvements. Choice of the data sets should be justified. The authors need to add a detailed comparative analysis of the proposed technique with existing methods.

Reviewer 1 ·

Basic reporting

The authors use Krawtchouk moments and Generalized
Pseudo-Zernike moments for the classification of fly wing
images and breast cancer mammograms. Experimental results reveal that the proposed feature extraction method produces prominent accuracy on both domains.

Experimental design

NA

Validity of the findings

The authors have produced some interesting results. However, I found the following lacking in the paper.
1. The performance has not compared with existing methods.
2. More evaluation methods may be used for the comparison
3. Additional papers related to breast cancer paper can enhance the quality of the paper:
Sitaula, C., Aryal, S. Fusion of whole and part features for the classification of histopathological image of breast tissue. Health Inf Sci Syst 8, 38 (2020). https://doi.org/10.1007/s13755-020-00131-7

Additional comments

The reviewer found lacking novelty in the paper. Specifically, the paper just utilizes well-established features without any novelty. Also, the performance comparison is insufficient. Rather than focussing on multiple datasets, I suggest working on only one with a deeper research understanding. I believe that the paper is unable to meet the expectation of the PeerJ Computer Science journal.

Reviewer 2 ·

Basic reporting

- English should be improved. There still have some unclear or ambiguous parts.
- Literature review should be re-organized to group some similar papers into one paragraph.
- Quality of the figures should be improved.

Experimental design

- The authors indeed had two case studies on fly wing images and breast cancer mammograms. Why did they use the title as 'a case study'? Also, the same in the whole text.
- The two case studies are also big questions. Why did they use these two datasets since they are not relevant to each other?
- DCOM format is at 2D or 3D?
- How did the authors deal with hyperparameter optimization of the models?
- Measurement metrics (i.e., accuracy, sensitivity, specificity, ...) have been used in previous biomedical studies such as PMID: 33816830, PMID: 33735760, and PMID: 33260643. Therefore, the authors are suggested to refer to more works in this description.
- Source codes should be provided for replicating the methods.

Validity of the findings

- In Figure 4, the text is not displayed clearly.
- Besides training, the authors should have some validation data.
- ROC curves and AUC values should be reported in binary classification.
- The authors should compare the predictive performance to previous studies on the same problem/data.

Additional comments

No comment.

Reviewer 3 ·

Basic reporting

The manuscript is clear and professional language is used.

Literature references and field background is sufficient.

Raw data is shared, results are relevant to the hypotheses.

Experimental design

The study is within the aims and scope of the journal.

Research question is well defined, relevant and meaningful. The research aims to fill the gap of predicting between fly species by their wing patterns and between benign or malignant masses in in mammograms. The study provides a model with a broad application area. The model is described with sufficient detail to replicate.

Validity of the findings

The limitations are clearly stated in the discussion. All data were provided, statistically sound and controlled. Benefit to the literature is stated, conslusions are linked to the scientific question at hand and limited to results.

---

## Round 0.2 · accepted · Accept

This work is novel in the sense that it examines how well orthogonal moments are able to produce good classification of benign and malignant states. The authors have sufficiently improved the manuscript in the light of reviewer's comments and provided adequate rebuttal. The revised version is in a good shape to be published in PeerJ.

Reviewer 1 ·

Basic reporting

NA

Experimental design

NA

Validity of the findings

Thank you for the rebuttal letter. However, I noticed my comments have not been addressed carefully.

1. If there is no novelty in the paper, it is worthless to publish the paper.
2. To publish the paper and disseminate the scientific claims, we need to evaluate thoroughly, for example, using more evaluation measures, etc.
3. We also need to compare our model with contending methods available in the literature although the dataset is new.

Given such questions unanswered sufficiently by the authors, the reviewer is inclined to clear rejection.

Additional comments

NA

Reviewer 2 ·

Basic reporting

No comment.

Experimental design

No comment.

Validity of the findings

No comment.

Additional comments

No comment.

Reviewer 3 ·

Basic reporting

No comment other than my previous comments.

Experimental design

No comment other than my previous comments.

Validity of the findings

No comment other than my previous comments.

---

## Author Rebuttal · Round 0.2

Rebuttal letter for manuscript #CS-2021:03:59010

19 June 2021

Dear editor,

We thank the reviewers for their time and comments. From the comments, we believe that our work may have been misunderstood. We have modified the title slightly (addition in italics) as "*On the* classification of simple and complex biological images using Krawtchouk moments and Generalized Pseudo-Zernike moments: a case study with fly wing images and breast cancer mammograms". In doing so, we wish to emphasise that this paper primarily discusses the applicability of orthogonal moments (using the Krawthchouk moments and the Generalized Pseudo-Zernike moments) in producing good classifications in biological problems, when they are used to extract features from relatively simple (represented using fly wing images), and relatively complex biological images (represented using breast cancer mammograms). It is not the intention of our work to evaluate a battery of potential feature extraction methods and potential classifier models in the analysis of these two kinds of biological images.

The following are our replies to the comments by all three reviewers.

Reply to Reviewer comments:

**Reviewer 1 (R1)**

1. R1: *Validity of the findings*

*"The authors have produced some interesting results. However, I found the following lacking in the paper.*

*1. The performance has not compared with existing methods.*

*2. More evaluation methods may be used for the comparison*

*3. Additional papers related to breast cancer paper can enhance the quality of the paper:*

*Sitaula, C., Aryal, S. Fusion of whole and part features for the classification of histopathological image of breast tissue. Health Inf Sci Syst 8, 38 (2020). https://doi.org/10.1007/s13755-020-00131-7"*

Authors: Regarding Item 1, we pointed out that while orthogonal moments may not be competitive compared to deep learning methods when used to analyse complex medical images (p.12, lines 381-393), they are nevertheless adequate for relatively simple biological images, such as fly wing images, which we show to be useful for fly species identification. For the fly wing image analysis problem, in pages 2-3 (line 83-115), we did describe a comparison of the result of classification using image features extracted from orthogonal moments with those using standard geometric morphometric data. There is no comparison in the literature for fly wing image analysis because published real data sets in this field are rare.

Additionally, the data set that we used is only very recently published (2020; https://doi.org/10.5061/dryad.95x69p8hf).

Regarding Item 2, we are unsure what is meant by "more evaluation methods may be used". If the reviewer means picking an optimal machine learning algorithm from a set of candidate algorithms, this is unnecessary in the present context as variation in choice of standard machine learning models has a much smaller effect on prediction performance compared to variation in choice of the more upstream feature extraction methods. In any case, the focus is on the effectiveness of orthogonal moments as the feature extraction method for which application of a suitable downstream machine learning method may yield meaningful results.

On Item 3, the suggested paper deals with prediction of breast cancer state based on histopathological images, which is entirely different from the use of predicting breast cancer state using mammograms.

2. R1: Comments for the Author

*"The reviewer found lacking novelty in the paper. Specifically, the paper just utilizes well-established features without any novelty."*

Authors: We do not claim novelty of methodology with respect to the use of orthogonal moments. However, our present work is novel in the sense that it re-examines how well orthogonal moments are able to produce good classification of benign and malignant states, using the CBIS-DDSM benchmark mammogram dataset. In doing so, we discover that previously reported performance (based on small test data set) using orthogonal moments are likely overly optimistic (pages 11 – 12; lines 367-379). We subsequently demonstrate that by considering the prediction output from mammogram image analysis (probability of malignancy) as a feature, using it in conjunction with the five expert features associated with the mammogram data (Models I to IV – pages 9-11; lines 322-348) enables reasonable prediction performance to be obtained. Concurrently, we also highlight the value of using orthogonal moments for the classification of fly species based on their fly wings (simpler biological images), for which we report encouraging results. We believe these results help researchers to keep orthogonal moments as an option when trying to do classification work with biological images, particularly when the problem studied does not yet have large sample size, and biological patterns in images are relatively simple.

3. R1: *"Also, the performance comparison is insufficient. Rather than focussing on multiple datasets, I suggest working on only one with a deeper research understanding."*

We disagree that the analysis of a single dataset using any machine learning methods, no matter how sophisticated, provides any "deeper research understanding". In fact, it is through analyses of multiple data sets that biases and limitations of particular feature extraction methods or machine learning methods become eventually clarified.

**Reviewer 2 (R2)**

1. R2: *Basic reporting*

*- English should be improved. There still have some unclear or ambiguous parts.*

*- Literature review should be re-organized to group some similar papers into one paragraph.*

*- Quality of the figures should be improved.*

Authors: We would appreciate if the reviewer can point to us specific sections of the manuscript where the quality of English language usage is doubtful. We reread our manuscript, and probably due to author cognitive bias, are unable to pinpoint the problematic parts.

Item 2 comes across us as vague to us and gives little information about how such change can make a difference to the main message that we want to bring out in the present work.

On Item 3, we agree that the quality of some of the figures does not yet satisfying PeerJ production level standards. However, they are generally intelligible and do not impede current understanding of the contents at the review stage.

2. R2: *Experimental design*

*" The authors indeed had two case studies on fly wing images and breast cancer mammograms. Why did they use the title as 'a case study'? Also, the same in the whole text."*

Authors: These two examples are not separate case studies, they are considered jointly in the manuscript to contrast the efficacy of orthogonal moments in extracting useful features that enable good prediction performance, in the case where its performance in relatively simple biological images (fly wings) is contrasted with that of relatively complex biological images (breast mammograms). It is entirely appropriate to consider these two data sets within the framework of a case study.

3. R2: *"The two case studies are also big questions. Why did they use these two datasets since they are not relevant to each other?"*

Authors: The focus of our work is to evaluate how well features extracted using orthogonal moments perform as far as the classification of biological images is concerned. These two data sets serve to illustrate the following two points: (i) the applicability of orthogonal moments in the classification of relatively simple biological images, as shown in the example of fly species prediction problem using fly wing images; (ii) the limitation in prediction performance when using orthogonal moments in the classification of complex biological images, such as prediction of benign and malignant states using mammograms.

By sharing our findings in the present work, we hope researchers would keep considering orthogonal moments as an option when doing classification work with biological images, especially in current times when researchers have a tendency to use deep learning methods to treat all image classification problems. Deep learning methods are unnecessary in the case of relatively simple biological images (as shown using the fly wing images), and not implementable when a problem has not yet have sufficiently large training sample size, or is still in an exploratory stage.

*4. R2: "DCOM format is at 2D or 3D?"*

Authors: DICOM is a communications protocol and a file format. It is a useful format that stores medical images data (e.g. mammogram, ultrasound, MRI images, etc.), together with information about the patient in a single file. For mammograms, the data is 2D. There is no need to handle the raw data in DICOM format as the owners of the CBIS-DDSM database have already converted the DICOM data files to PNG file format. To make this clearer, we added the following sentence on page 4 (end of line 165).

"The images were downloaded from the CBIS-DDSM database in the PNG file format."

*5. R2: "How did the authors deal with hyperparameter optimization of the models?"*

Authors: For the hyperparameters of GUIDE decision trees, the standard k=0.5 parameter produces trees that are neither too complex nor too simplistic. The kernel discriminant model applied at the partitioned data spaces at the tree terminals is essentially data-optimised because it is based on the maximum-likelihood method. The default GUIDE random forest hyperparameters with user-specified 2001 trees produces reasonably good results. Generally, there is a broad range of hyperparameter values which can be used, and they should produce more or less similar prediction performance.

We refer the reviewer to page 7 (line 267-272).

"For classification, we used a kernel discriminant model in the GUIDE (Generalized, Unbiased, Interaction Detection and Estimation) classification and regression tree program (Loh, 2009; Loh, 2014). The kernel method is a non-parametric method that estimates a Gaussian kernel density (Silverman, 1986) for each class in a node, and uses the estimated densities in a maximum likelihood framework for classification. The tree complexity parameter k-SE in GUIDE was set at the default value of 0.5, with the number of cross-validated trees set at 10."

*6. R2: Measurement metrics (i.e., accuracy, sensitivity, specificity, ...) have been used in previous biomedical studies such as PMID: 33816830, PMID: 33735760, and PMID: 33260643. Therefore, the authors are suggested to refer to more works in this description.*

Authors: For the breast cancer mammogram data, there are only two classes, in approximately equal proportions. So the use of accuracy is appropriate, as is sensitivity and specificity. Sensitivity and specificity are not useful metrics in the fly image problem, as there are 15 classes. For this problem, only the accuracy metric was used, and to account for estimation variance probabilistically, a 95% Bayesian credible interval was provided to this metric.

*7. R2: Source codes should be provided for replicating the methods.*

Authors: We already provided the source codes in the initial submission as supplemental files. This is corroborated by comments from Reviewer 3.

8. R2: Validity of the findings

*"- In Figure 4, the text is not displayed clearly.*

*- Besides training, the authors should have some validation data.*

*- ROC curves and AUC values should be reported in binary classification.*

*- The authors should compare the predictive performance to previous studies on the same problem/data."*

Authors: Item 1: The texts in the figure are class labels. They are "not clear" because they overlap substantially. We have tried adjusting the font size but find the present one is the best.

Item 2: We refer to page 7 (lines 273-277):

"For Problem 1, it was not feasible to split the samples into another test set for assessing generalization error, because the average sample size per class was already small (about 5). Hence, we applied the random forests ensemble classifier (2001 trees) and obtained the out-of-bag error estimate for generalization error. For the CBIS-DDSM data set in Problem 2, we randomly chose 70% of the data set for training, and the remainder 30% for testing."

It was not feasible to split the data into a test set in Problem 1, as the sample sizes of each species are already small. For this reason, the random forests model with out-of-bag error estimate for generalization error was used. This is a reasonable substitute for estimating prediction error when sample size is not large. For Problem 2, 30% of data was used for testing.

Item 3: We understand that ROC curve is used so often in evaluation of model performance that, somehow, it is expected to be presented in every binary classification problem. However, we believe that ROC curve should only be used when it can be interpreted in the context of the problem where it is used. In Problem 2 which involves breast cancer mammograms, the ROC curve and the resultant AUC metric are unsuitable because the BI-RADS feature is

considered in the analysis (Model I and Model II (page 11; Table 1, Table 2). Jiang & Metz (2010) show that the BI-RADS feature is inappropriate for construction of ROC analysis as the encodings used in the BI-RADS feature have context-specific meaning in radiology. In addition, the problem of whether AUC of the ROC curve means anything to the subject matter experts (clinicians) in a radiology setting is also well-documented by Halligan et al. (2015). We quote from them:

*"Sensitivity and specificity are familiar concepts to clinicians, who are used to interpreting the results of diagnostic tests in these terms. In contrast, ROC AUC means little to clinicians (especially non-radiologists), patients, or health care providers. While a test whose AUC is 0.9 is considered "better" than one of 0.8, what does this mean for patients and what is clinically important? It is well established that diagnostic tests are understood best when presented in terms of gains and losses to individual patients [11]. AUC lacks clinical interpretability because it does not reflect this. Clinicians are uninterested in performance across all thresholds - they focus on clinically relevant thresholds. However, because AUC measures performance over all thresholds, it includes both those clinically relevant and clinically illogical. Moreover, different tests can have identical AUC but different performance at clinically important thresholds.*"

References:

1.Halligan, S., Altman, D.G. & Mallett, S. (2015). Disadvantages of using the area under the receiver operating characteristic curve to assess imaging tests: A discussion and proposal for an alternative approach. *European Radiology*, 25: 932-939.

2. Jiang, Y. & Metz, C.E. (2010). BI-RADS data should not be used to estimate ROC curves. *Radiology*, 256: 29-31.

Item 4: For the breast cancer mammograms, such comparisons are pointed out in page 11-12 (lines 364-392). For the fly wing images, the present application of orthogonal moments to analysis of fly wing images is novel because the data set used is very recent (2020; https://doi.org/10.5061/dryad.95x69p8hf). We actually reported a comparison of the result of classification using image features extracted from orthogonal moments with those using geometric morphometric data (page 2, line 83-115; page 8, line 311-322).

**Reviewer 3 (R3)**

*1.R3 "Basic reporting*

*The manuscript is clear and professional language is used. Literature references and field background is sufficient. Raw data is shared, results are relevant to the hypotheses.*

*Experimental design*

*The study is within the aims and scope of the journal. Research question is well defined, relevant and meaningful. The research aims to fill the gap of predicting between fly species by their wing patterns and between benign or malignant masses in in mammograms. The study provides a model with a broad application area. The model is described with sufficient detail to replicate.*

*Validity of the findings*

*The limitations are clearly stated in the discussion. All data were provided, statistically sound and controlled. Benefit to the literature is stated, conslusions are linked to the scientific question at hand and limited to results."*

Authors: We thank the reviewer for positive opinions about our work.